# Basic Activity of Daily Living Evaluation of Children with Autism Spectrum Disorder: Do-Eat Washy Adaption Preliminary Psychometric Characteristics

**DOI:** 10.3390/children10030514

**Published:** 2023-03-05

**Authors:** Hana Levy-Dayan, Naomi Josman, Sara Rosenblum

**Affiliations:** 1Ministry of Education, Northern District, 15 Palyam Blvd, Haifa 33095, Israel; 2The Laboratory of Complex Human Activity and Participation (CHAP), Department of Occupational Therapy, Faculty of Social Welfare & Health Sciences, University of Haifa, Mount Carmel, Haifa 3498838, Israel

**Keywords:** autism spectrum disorder, basic activities of daily living, executive function, sensorimotor, performance, assessment

## Abstract

This preliminary study aims to demonstrate the reliability and validity of the adapted Do-Eat Basic activities of daily living (BADL) assessment for children with autism spectrum disorder (ASD). Participants were 53 children ages 6–10 years: 17 diagnosed with high-severity ASD (HS-ASD) and 16 with low-severity ASD according to the DSM-5 and based on the CARS-2, and 20 controls with typical development. Measurement tools were the adapted Do-Eat Washy (Washy), Participation in Childhood Occupations Questionnaire (PICO-Q), and Pediatric Evaluation of Disability Inventory (PEDI). The Washy domains exhibited high internal consistency (α = 0.841–0.856). Significant differences were found between the HS-ASD and other groups in the Washy domains, exhibiting discriminant validity. The Washy convergent and concurrent validity indicated good results. A highly substantial negative correlation was shown between the Washy and three PICO-Q ADL difficulty-in-performance items (*p* < 0.001): bathing (−0.550), hygiene (−0.571), and handwashing (−0.733). The Washy performance scores and the PEDI total score demonstrated a strong correlation. (*r* = 0.799, *p* < 0.001). Primary results indicate that, following further research on larger representative samples, the Washy may be a reliable and valid tool for assessing BADL among children with ASD.

## 1. Introduction

Autism spectrum disorder (ASD) is a developmental disorder that appears at an early age and has lifelong effects. It is defined by impaired social communication skills and restricted, repetitive patterns of behavior, interests, and activities. According to the current definition in the *Diagnostic and Statistical Manual of Mental Disorders* (5th ed.; DSM-5) [1], the clinical picture of ASD is divided into three functioning levels: high severity (HS) Level 3, moderate severity (Level 2), and Level 1 low severity (LS). These severity levels focus on how much support children need in their daily lives..

Researchers have shown that many children with ASD suffer from daily functioning difficulties and require extensive support and assistance in various functional domains in different contexts, including Activities of daily living (ADL) [2].

ADL comprise a broad spectrum of daily activities divided into basic and instrumental ADL. This study focuses on basic ADL (BADL), which consists of tasks people perform during the day for self-care, such as hygiene, feeding, bathing, dressing, and grooming [3]. In comparison to other areas, children with ASD require a higher amount of support with ADLs., even though adults support many children in performing these tasks [4].

Self-care dependence is a significant problem among children with ASD, with over 50% of young children with ASD unable to perform self-care independently [5,6,7].

However, despite the well-known relationships between their self-care abilities and reduced participation in various domains, such as leisure, chores, and community mobility [8,9,10,11], literature on self-care characteristics of school-aged children with ASD remains scarce.

Several of the existing studies demonstrated significantly lower self-care performance abilities among children ages 4 to 10 years with ASD compared to those of children with typical development [12,13]. Those studies used parents’ questionnaires, such as the Pediatric Evaluation of Disability Inventory (PEDI) [14], the Self-Assessed Activities of Daily Living Scale (SAADL) [15], and the Vineland Adaptive Behavior Scales [16]. The Activities of Daily Living Evaluation (ADL-E) specifically measured both BADL (eating, personal hygiene, and getting dressed) and executive functions (EF) [17] among school children between 6 and 12 years old. However, Caregiver evaluation scales may produce inaccurate results because they cannot evaluate actual self-care performance abilities [6,18,19].

Among studies using performance-based assessments, the observational Assessment of Motor and Process Skills (AMPS) [20] was utilized to assess ADL performance, tied to motor and process skills simultaneously. The AMPS, however, can only be administered by occupational therapists who have successfully completed a 5-day training course, and it is insufficient on its own to test children’s ADL performance [12]. Despite the link between self-care abilities and participation of children with ASD [21], there is a lack of accessible performance-based BADL evaluations that include self-care performance tasks for school-aged children with ASD. Hence, in order to provide a complete picture of these children’s self-care performance, an observational measure in addition to a caregiver rating scale is required. Taking into account both performance-based exams and parent questionnaires might provide more valuable knowledge regarding their everyday skills..

Besides performance-based assessments for real daily self-care activity, assessments are also needed which simultaneously capture the sensorimotor (SM) and EF abilities tied to ADL performance [22,23]. It is estimated that over 90% of children with ASD show symptoms of sensory difficulties [24], currently included in the latest definition of ASD (*DSM-5*) [1] as a diagnostic characteristic. Deficient fine and gross-motor skills were found to relate to various difficulties in self-care functioning, including bathing and eating independently, among children with ASD [25,26]. Ahmed et al. [27] used the Barthel Index Scale to determine that most of the 60 children (aged 4 to 15 years) with ASD in their study had sensorimotor problems tied to their ADL performance. This finding supports Whyatt and Craig’s study [28], demonstrating the fundamentality of developmental SM difficulties as an ASD core-element symptom. Furthermore, such impairments can adversely affect children’s participation and performance in ADL activities [21].

Alongside SM skills, daily functioning leans on EF [29,30]. High-level cognitive processes known as executive functions (EF) include planning, working memory, impulse control, inhibition, and shifting, that contribute to purpose, strategy, and goal-directed actions [31,32]. Panerai et al.’s [33] study of EF in ASD indicated that executive dysfunctions in planning, flexibility, and behavior regulation in ecological contexts seemed common to the ASD continuum (with and without intellectual disability). Gardiner and Iarocci [34] investigated the relative impact of daily EF as assessed by parent reports across a variety of functional outcomes, including ADL.. They suggested that EF relates to motor impairments in children. Sensory difficulties and EF may affect personal care skills. These activities entail multiple sensations, including tactile sensations, such as handling soap, towels, and brushes. Children with autism may not correctly process the sensory information they receive from their bodies and the environment during self-care. Activities that must be completed in a set sequence, for example, applying shampoo before rinsing, must be understood and remembered.

The Do-Eat assessment was developed to evaluate daily task skills in children with neurodevelopmental disabilities [35,36,37], such as developmental coordination disorders (DCD) [38], attention deficit hyperactive disorders (ADHD) [39] and ASD [40] As an ecological test, The Do-Eat is used to assess areas of instrumental ADLs’ strength and difficulty in the children’s natural environment. [41] Three activities are given to the children: making a sandwich, making chocolate milk, and filling out a certificate of excellence for oneself. All through their performance, they receive (1) a performance score, (2) a score for SM skills analysis, and (3) a score for EF skills analysis. Do-Eat interrater reliability ranges from 0.92 to 1.00 [36], and high internal consistency has been found for each group of items (α = 0.89–0.93). In addition, Do-Eat demonstrated good content and discriminant validity between gender and age groups and between children with DCD and typically developing children aged 5.0 to 6.5 years, *t*(57) = −11.94, *p* < 0.01) [36,37,38].

Given the need for extensive examination of children’s actual performance along with their EF and SM abilities, and following Katz Zetler et al.’s study results [40] which reiterated the importance of further adapting the Do-Eat to children with ASD with varied functional abilities levels, we adapted the Do-Eat [36,37] into the Do-Eat Washing Hands Structured Observation (Washy) [42]. We adjusted it for ASD and used a BADL self-care task to adapt the tool to children with low functioning levels. To use a task performed on a routine, daily basis and in home and school settings, and that can be done in a public space so that the child’s privacy is not compromised, we chose handwashing. The Washy dynamic assessment circumvents challenges that often surface when evaluating this population [43].

Thus, the purpose of the current study was to demonstrate the reliability and validity of the Do-Eat Washy assessment. The research hypotheses were: **HA1.** *The internal consistency of the Washy assessment will be at a Cronbach alpha level equal to or greater than 0.70.*
**HA2.** *Significant differences will be found among children with HS-ASD, LS-ASD, and typical development in the Washy assessment performance criteria, thus establishing discriminate validity.*
**HA3.** *Significant correlations will be found among the four Participation in Childhood Occupations Questionnaire (PICO-Q) ADL items, which involve performing a sequence of actions (bathing, hygiene, clothing, and washing hands), and the Washy assessment performance scores, thus establishing its convergent validity.*
**HA4.** *Significant correlations will be found among the Washy performance and degree of independence scores and the Pediatric Evaluation of Disability Inventory (PEDI) independence subtest in self-care, thus establishing concurrent validity.*


## 2. Materials and Methods

### 2.1. Participants

The Ethics Committee of the Faculty of Social Welfare and Health Sciences at the University of Haifa (No. 271/17) and the Israeli Ministry of Education (No. 9610) provided ethical approval for this study. This study used a convenience sample of 53 children aged 6 to 10 years (M = 7.75 years, SD = 1.06). All participants were Hebrew speakers, most of them Jewish, who attended Northern Israel district schools. Participants with ASD were recruited by parent referral from special education schools, special education classes in normal schools, or regular schools. The diagnoses were made by pediatricians or clinical psychologists according to DSM-5 criteria [1]. The autism function level was verified according to the Childhood Autism Rating Scale (2nd ed.; CARS-2; [44]) completed by parents. Twenty children without a history of developmental or psychiatric disorders were recruited from regular schools through parent referral to constitute the control group. As presented in Table 1, there were no significant group variations in the participants’ age, gender, socioeconomic level, or religion. However, a significant difference was found between the mothers’ education in the research and control groups; thus, this variable was held as constant in the statistical analyses.

### 2.2. Materials and Design

#### 2.2.1. Do-Eat Washy

The Do-Eat Washing Hands Structured Observation (Washy) [42] was developed based on the Do-Eat Performance-Based Assessment for Children (Do-Eat) [36] and adjusted for ASD [40]. A BADL self-care task was used to adapt the tool for children with low functioning levels. Because it was important to use a task performed on a routine, daily basis, we chose handwashing. The Washy includes the same four domains as the Do-Eat [36]: (a) a task execution analysis (10 items), (b) SM skills (six items), (c) EF (nine items), and (d) 10 distinctive performance features. For the structure and outcomes measures, part of the Do-Eat was adapted for children with ASD. Specifically, a series of photographs was constructed to illustrate the task sequence, enabling the independent performance of the task. The child is asked to perform a handwashing task and, throughout their performance, receives (1) a task execution score, (2) SM skills analysis score and (3) EF skills analysis score. The child’s performance receives a total score ranging from 10 (not independent) to 60 (independent) and two component scores. One component score for independent performance on each task is from 0 (not independent) to 1 (independent), and the range of the overall component score is 0 (not independent) to 10 (Fully independent). The second component scores the amount of assistance provided during each performance level from 10 (with full assistance) to 50 (without assistance).

The SM skills include six items: motor function, posture and movement, motor planning, bilateral coordination, fine-motor coordination, and sensation. Based on the child’s performance, these are scored from 1 (unsatisfactory) to 5 (very good) and include the average total score. The nine EF analysis components are attention, initiation, sequencing, transitioning from one activity to another, spatial organization, temporal organization, inhibition, problem-solving, and remembering instructions. Similar to the SM analysis, the EF analysis overall score ranges from 1 (unsatisfactory) to 5 (very good), with an average total score. Scores for the number of characteristics viewed range from 0 (not observed) to 10 (observed all).

The Washy content was validated by eight experienced occupational therapists who are experts in ASD. This procedure was used to determine the degree of correspondence between the tool scenario and the consistency of content and face validity. A 100% agreement for performance and assistance items was achieved. Further reliability and validity were established in the current study. High internal reliability was found for the Do-Eat Washy assessment performance items (α = 0.86) and SM skills and EF criteria (α > 0.84). See Figure 1 for an overview of the Washy. See Appendix A for the Washy assessment (English translation).

#### 2.2.2. CARS-2

The CARS-2 [44] is a behavioral rating scale used for assessing the presence and severity of ASD symptoms. We verified the participants’ functioning levels using the CARS-2 standard form to assess the HS-ASD study group. We used the high-function form to assess the LS-ASD study group. Both scales consist of 15 items rated on a 4-point scale relative to the extent to which the behavior is differentiated from others of the respondent’s age. Each questionnaire has three cross-sections: not autism, mild–moderate autism, and severe autism. The CARS-2 has demonstrated good internal consistency, interrater reliability, and validity (0.93) [44,45]. Scores from 37 to 60 (standard form) indicate HS-ASD; the HS-ASD group in this study ranged from 37 to 47 (M = 41.7, SD = 3.47). Scores between 28 and 33.5 (high-function form) indicate LS-ASD; the LS-ASD group in this study ranged from 28 to 33.5 (M = 30.15, SD = 1.73).

#### 2.2.3. PICO-Q ASD

The PICO-Q version for ASD [46] is a 34-item scale measuring the performance level, enjoyment, and frequency of children’s participation in daily activities in five domains: ADL (16 items), school activity (seven items), play and leisure (four items), social functioning (four items), and routines and habits [46]. Each participant in this study had the option of selecting two additional activities, and a handwashing activity was added. Each item is scored on its four components: performance difficulty, frequency, level of involvement, and enjoyment, with scores ranging from 1 (low) to 5 (high). In Heller’s [47] study, reliability and validity were established among 23 parents of children with ASD. In the current study, internal consistency coefficients for frequency and involvement across the family, school, and community were 0.63, 0.42, and 0.57, respectively [42].

#### 2.2.4. PEDI Self-Care Subtest

The PEDI was established as a parent report about children’s ADL performance abilities [14]. The self-care subtest includes eight items that measure the typical amount of assistance provided by the caregiver with tasks such as personal hygiene and hand washing. Each item is scored on a scale from 5 (independence; no physical help or supervision) to 0 (total help; caregiver does the entire task) and includes a rating of the frequency of adjustments. The total scale score ranges from 0 to 100, with lower scores indicating higher severity. The PEDI internal consistency coefficients were high (>0.09) [14], and a similar value (0.93) was found in the current study.

### 2.3. Procedure

Caregivers completed the PICO-Q ASD [46] and PEDI self-care subtest [14], and the children performed the Washy [40].

### 2.4. Analysis

We used IBM SPSS (Version 24) for data analysis, Cronbach’s alpha to assess internal reliability (Hypothesis A1), and analysis of variance (ANOVA) and multivariate ANOVA to establish discriminate validity for the Washy performance criteria among children with HS-ASD, children with LS-ASD, and children with typical development (Hypothesis A2). Spearman’s correlation tests were conducted between the performance scores for ADL tasks involving a sequence in the PICO questionnaire (bathing, hygiene, clothing, handwashing) and the Washy assessment scores (performance, SM, and EF) to establish convergent validity (Hypothesis A3). Finally, Pearson and Spearman’s correlations were made between the Washy assessment performance scores and the parents’ evaluations of the degree of independence in the Washy and PEDI to test the Washy assessment’s corresponding validity (Hypothesis A4).

## 3. Results

### 3.1. Hypothesis A1: Washy Assessment Internal Consistency

The internal consistency coefficients were 0.856 for the Washy performance score, 0.843 for SM skills, and 0.84 for EF. This means good alignment between the Washy component scores.

### 3.2. Hypothesis A2: Washy Discriminate Validity

The coefficients were 0.92 for total SM and EF skills. A significant difference was found between the group of children with HS-ASD and the other groups in the performance criteria of the Washy assessment, F(2,49) = 66.77, *p* <. 001, η*p*^2^ = 0.73, and SM and EF skills, F(4,96) = 22.97, *p* < 0.001, η*p*^2^ = 0.49. In addition, significant between-group differences were found in the number of special characteristics observed, *F*(2,49) = 17.91, *p* < 0.001, η*p*^2^ = 0.42). These results establish a discriminate validity for the Washy between children with HS-ASD, LS-ASD, and typical development. Table 2 presents the Washy’s averages, standard deviations, and F scores.

### 3.3. Hypothesis A3: Washy Convergent Validity

A significant high negative correlation was found among the Washy assessment scores (performance, SM, and EF) and three PICO-Q ADL difficulty-in-performance items (*p* < 0.001): bathing (−0.55), hygiene (−0.57), and washing hands (−0.73), and a moderate negative correlation with the clothing item (−0.39, *p* < 0.01) in the Washy (Figure 2). This confirms Hypothesis A3 and contributes to establishing convergent validity for the Washy assessment.

### 3.4. Hypothesis A4: Washy Concurrent Validity

The performance scores of the Washy and the PEDI were shown to be significantly highly correlated (*r* = 0.80, *p* < 0.001) for all groups, while a moderate but substantial association was found at the HS-ASD group (*r* = 0.50, *p* < 0.01). In addition, a significant high correlation was found between the degree of independence in the Washy (based on parent reports) and the PEDI subtest of independence in self-care (*r* = −0.76, *p* < 0.001). Hence, the higher the parents’ report score (poorer independence), the lower the PEDI score. This confirms Hypotheses A4 and contributes to establishing the concurrent validity of the Washy assessment.

## 4. Discussion

This study’s goals were to present the Do-Eat Washy development and the establishing of its reliability and validity. The findings supported the hypotheses. By looking at internal consistency as well as discriminant, convergent, and concurrent validity, we were able to establish the Washy reliability. The internal consistency analysis revealed that the items evaluated in each test and questionnaire category had a high degree of correlation. This significant correlation demonstrates that each category’s items evaluate the same topic [48].

This study helps provide a picture of daily function, difficulty, and dependence in ADL tasks for children with HS-ASD compared to children with LS-ASD and with typical development. The results indicate increased performance according to the severity level. That is, the HS-ASD group demonstrated greater difficulty in ADL and greater dependence on parents and caregivers. The study also provides information on the participation of children with autism by combining parental reporting instruments with a performance-based instrument. The preliminary results validate the Washy assessment tool, enabling a valid and reliable performance-based tool to assess the functioning of children with HS-ASD and measure their ADL skills performance relative to their SM and EF skills in the context of their everyday lives. These findings, though, should be viewed in light of the limitations of this study. Data were gathered from caregivers who tended to come from similar socioeconomic backgrounds, which might have biased the sample. Although the study includes direct observation of the children’s participation, as measured by the Washy, only one ADL area was observed. Moreover, there is a discrepancy between the Washy assessment and the needs and characteristics of children with LS-ASD and typical development. These preliminary results indicate the validity and reliability of the Washy tool, but larger and more representative studies are required to continue to establish it as a performance-based tool among this population. The findings highlight daily functioning and SM and EF skills as important aims for possible intervention for children with HS-ASD. Sensorimotor impairments impact the autonomy of children with ASD; thus, interventions should aim to improve and support the development of their SM skills [6]. This study may increase awareness of the participation difficulties that children with ASD face in various areas of life. We encourage the development of additional assessment tools and effective intervention programs suited to this population’s specific difficulties and considering their functioning in their natural environment. Thus, the Washy was preliminarily found to be a sensitive tool for assessing children with HS-ASD. Practitioners and stakeholders could use the Washy when developing and implementing interventions to address ADL difficulties in individuals with ASD. The scoring can indicate the amount of support needed, the current performance level, and the point of difficulty for the EF and SM. The function of the home and school environment might be evaluated using the Washy. It may also be used to evaluate how well children are performing both before and after the intervention to enhance self-care abilities such as hand washing and to gain a better understanding of the children’s EF and SM.

## 5. Conclusions

The study provides information on the participation of children with autism utilizing parental reporting tools in conjunction with a performance-based tool. Children vary in their difficulty (PICO-Q) in participation according to the autism severity level. Higher severity levels show greater difficulty and parental dependence in ADL. The Washy had high internal consistency (α = 0.84–0.85). Preliminarily, Washy is reliable and valid performance-based assessment for BADL functioning. Further, daily functioning and SM and EF skills are important considerations for professionals. the Washy could be used to assess function at home and school environment. Moreover, it can be used before and during the intervention to improve self-care skills such as washing hands.

## Figures and Tables

**Figure 1 children-10-00514-f001:**
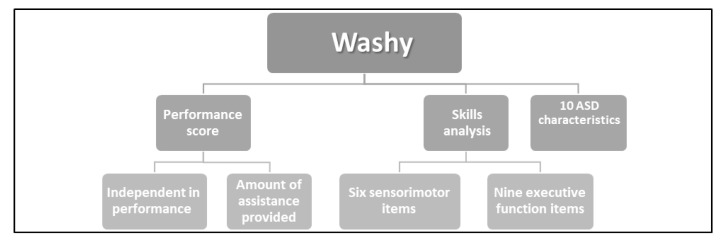
The Do-Eat Washing Hands Structured Observation (Washy).

**Figure 2 children-10-00514-f002:**
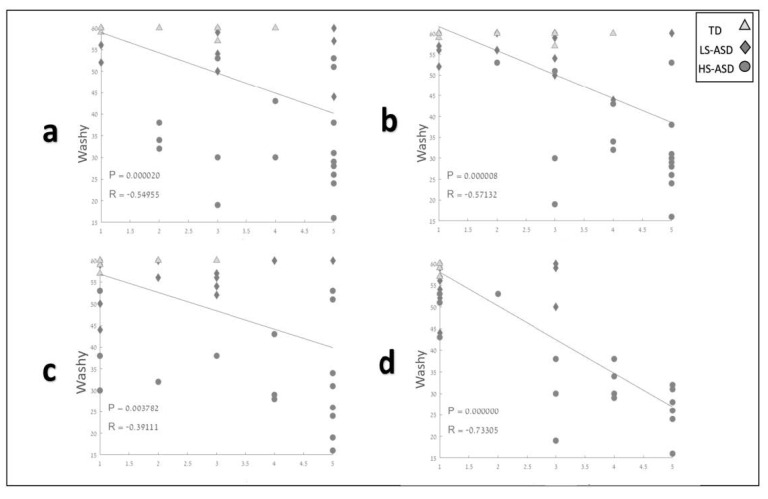
Correlations between Washy and Participation in Childhood Occupation (PICO-Q) performance measures across experiment groups. Note. The relationship between Washy and PICO-Q scores is illustrated using Spearman’s rank correlation coefficient tests of individuals’ Washy’s performance score against their PICO-Q difficulty scores item, including (**a**) PICO-Q-1, (**b**), PICO-Q-2, (**c**) PICO-Q-4, (**d**) and PICO-Q-16. The participants with typical development (TD; triangles) usually received the lowest scores in both assessments, followed by participants with LS-ASD (diamonds), and ending with participants with HS-ASD (circles), who received the highest (worst) scores in both assessments.

**Table 1 children-10-00514-t001:** Demographic characteristics by group.

Variable	HS (*n* = 17)	LS (*n* = 16)	Typical (*n* = 20)	F/P
	*M* (*SD*)			
Child’s age	7.52 (0.83)	8.14 (0.94)	7.62 (1.26)	1.68
Mother’s education	14.52 (2.06)	14.81 (2.34)	17.05 (2.01)	** 7.84
Father’s education	14.25 (2.79)	15.64 (3.29)	16.15 (1.53)	2.58
Child’s gender				
Male	12.00 (70.60)	11.00 (68.80)	13.00 (65.00)	
Female	5.00 (29.40)	5.00 (31.20)	7.00 (35.00)	*ns*
Socioeconomic level				
Low	0	1.00 (6.30)	1.00 (5.00)	
Average	13.00 (76.50)	9.00 (56.20)	15.00 (75.00)	
Above average	4.00 (23.50)	6.00 (37.50)	4.00 (20.00)	*ns*
Religion				
Judaism	15.00 (88.20)	13.00 (81.30)	20.00 (100.00)	
Christianity	0	2.00 (12.50)	0	
Other	2.00 (11.80)	1.00 (6.30)	0	*ns*
School class				
First grade	7.00 (41.20)	6.00 (37.50)	8.00 (40.00)	
Second grade	7.00 (41.20)	2.00 (12.50)	4.00 (20.00)	
Third grade	2.00 (11.80)	8.00 (50.00)	6.00 (30.00)	
Fourth grade	1.00 (5.80)	0	2.00 (10.00)	*ns*
ASD in family?				
Yes	2.00 (11.80)	1.00 (6.25)	1.00 (5.00)	
No	15.00 (88.20)	15.00 (93.75)	19.00 (95.00)	*ns*

Note. HS = high severity autism spectrum disorder (ASD) study group; LS = low severity ASD study group; Typical = control group with typical development; ns = not significant. ** *p* < 0.01.

**Table 2 children-10-00514-t002:** Washing Hands Structured Observation (Washy) scores: averages, standard deviations, and F values.

Variable	HS-ASD (*N* = 17)	LS-ASD (*N* = 16)	Typical (*N* = 20)	*F*(2,49)	η*p²*
	*M* (*SD*)				
Washy performance					
Total score	33.82 (11.01)	56.68 (4.64)	59.80 (0.69)	*** 66.77	0.73
Independence	5.17 (1.87)	9.31 (1.01)	10.00	*** 72.66	0.75
Assistance	28.64 (9.47)	47.37 (3.98)	49.80 (0.69)	*** 59.94	0.71
Washy skills					
Sensorimotor	2.82 (0.85)	4.49 (0.43)	4.83 (0.18)	*** 55.76	0.69
Executive function	3.17 (0.73)	4.55 (0.33)	4.86 (0.21)	*** 55.38	0.69
No. characteristics	1.88 (0.92)	1.00 (0.96)	0.05 (0.22)	*** 17.91	0.42

Note. *** *p* < 0.001.

## Data Availability

The datasets used and/or analyzed during the current study are available from the corresponding author on reasonable request.

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
