# Peer review of "Basic Activity of Daily Living Evaluation of Children with Autism Spectrum Disorder: Do-Eat Washy Adaption Preliminary Psychometric Characteristics"

_children, 2023, doi:10.3390/children10030514_

Round 1

Reviewer 1 Report

This work is important in the effort to further define, categorize, and provide appropriate interventions for children with autism spectrum disorder.

I suggest combining the second paragraph with the first paragraph. At the end of the first paragraph, Level 3 is defined as high severity and Level 1 is defined as low severity--the sentence should include Level 2 and how it is defined, such as "moderate" severity. This can eliminate the first sentence of the next paragraph that is in part redundant. Then: "These severity levels focus around how much support children need in their daily lives." And then the last sentence as written "Researchers have shown..."

Line #49 should follow and be part of the previous one-sentence paragraph.

The literature review is very strong. A sample size of 53 children in this study is adequate. The 4 hypotheses are clear. Figure 1 does a very good job of anchoring the reader in the complex explanation of the range of items covered: performance score, skills analysis, and 10 ASD characteristics.

The authors could do more to ensure that those without statistical expertise can understand the major findings of their work. For example, in Section 3 Results, under 3.1 the one sentence reporting internal consistency coefficients should be followed by a second sentence that could begin: "This means...."

The Discussion section confirms that the results support all 4 hypotheses. I think the Discussion and Conclusion sections could be changed a bit for clarity and could further underscore the importance of this work. I never like to end an article with the limitations, and prefer to end on a positive note that gives ideas of how this work can be utilized. I would put line #s 318-326 at the end of the Discussion section. Line #s 288-292 are better placed in the Conclusion. Line #s 302-309 are also better when placed in the Conclusion. Early childhood educators and human service professionals are interested in this work and so some recommendations that may be helpful to them are in order here. For example, the authors need to provide some examples of where and when the Washy could be used.

The current title is too cumbersome. Potential alternative title for the authors to consider: Basic Activity of Daily Living Evaluation of Children with Autism Spectrum Disorder: The Do-Eat Washy Adaptation

Author Response

Anastasia Yu

Editorial Board Member

Journal Children

Dear Editorial Board,

Enclosed for your review is our revised article entitled: “Basic Activity of Daily living (BADL) evaluation among children with ASD: Preliminary psychometric properties of the Do-Eat Washy adaptation.”

We thank the editors and the reviewers for their comments, which allowed us to improve the paper. Each comment is addressed in the attachment, and major content changes are highlighted in red in the revised manuscript.

Sincerely,

Hana Levy-Dayan, Prof. Naomi Josman and Prof. Sara Rosenblum

Reviewer 2 Report

The paper presents a study on activities of daily living among children with Autism Spectrum Disorder. It focuses on psychometric properties of the Do-Eat Washy adaptation.

The research questions are relevant for both research and practice. The manuscript is well written.

Important strengths include the detailed assessments.

However, the sample of 53 children is very small in the research context. Has an a priori power analysis been conducted? Are estimates reliable with such a small sample?

Were differences in psychometric properties according to disease severity, sex, education level, etc.?

The conceptual rationale could be elaborated in more detail in the Introduction and the Discussion.

The practical relevance could be illustrated with more detailed examples from everyday life.

Please avoid abbreviations in the title.

Author Response

(The authors gave the same response as above.)
